# Fracture Mechanics and Oxygen Gas Barrier Properties of Al_2_O_3_/ZnO Nanolaminates on PET Deposited by Atomic Layer Deposition

**DOI:** 10.3390/nano9010088

**Published:** 2019-01-11

**Authors:** Vipin Chawla, Mikko Ruoho, Matthieu Weber, Adib Abou Chaaya, Aidan A. Taylor, Christophe Charmette, Philippe Miele, Mikhael Bechelany, Johann Michler, Ivo Utke

**Affiliations:** 1Mechanics of Materials and Nanostructures Laboratory, Empa- Swiss Federal Laboratories for Materials Science and Technology, Feuerwerkerstrasse 39, CH-3602 Thun, Switzerland; vipin.phy@gmail.com (V.C.); mikko.ruoho@empa.ch (M.R.); Johann.Michler@empa.ch (J.M.); 2CSIR-Central Scientific Instruments Organisation, Sector 30, Chandigarh 160030, India; 3Institut Européen des Membranes, IEM, UMR-5635, Univ Montpellier, CNRS, ENSCM, 4095 Montpellier, France; matthieu.weber@umontpellier.fr (M.W.); adib.a.c@hotmail.com (A.A.C.); christophe.charmette@umontpellier.fr (C.C.); Philippe.Miele@umontpellier.fr (P.M.); mikhael.bechelany@umontpellier.fr (M.B.); 4Materials Department, University of California, Santa Barbara, CA 93106, USA; aidantaylor@ucsb.edu; 5Institut Universitaire de France, 1 rue Descartes, 75231 Paris, France

**Keywords:** atomic layer deposition, mechanics, gas barrier, tensile strain testing, adhesion, sub-surface growth, failure analysis, delamination, thin film, Al_2_O_3_, ZnO

## Abstract

Rapid progress in the performance of organic devices has increased the demand for advances in the technology of thin-film permeation barriers and understanding the failure mechanisms of these material systems. Herein, we report the extensive study of mechanical and gas barrier properties of Al_2_O_3_/ZnO nanolaminate films prepared on organic substrates by atomic layer deposition (ALD). Nanolaminates of Al_2_O_3_/ZnO and single compound films of around 250 nm thickness were deposited on polyethylene terephthalate (PET) foils by ALD at 90 °C using trimethylaluminium (TMA) and diethylzinc (DEZ) as precursors and H_2_O as the co-reactant. STEM analysis of the nanolaminate structure revealed that steady-state film growth on PET is achieved after about 60 ALD cycles. Uniaxial tensile strain experiments revealed superior fracture and adhesive properties of single ZnO films versus the single Al_2_O_3_ film, as well as versus their nanolaminates. The superior mechanical performance of ZnO was linked to the absence of a roughly 500 to 900 nm thick sub-surface growth observed for single Al_2_O_3_ films as well as for the nanolaminates starting with an Al_2_O_3_ initial layer on PET. In contrast, the gas permeability of the nanolaminate coatings on PET was measured to be 9.4 × 10^−3^ O_2_ cm^3^ m^−2^ day^−1^. This is an order of magnitude less than their constituting single oxides, which opens prospects for their applications as gas barrier layers for organic electronics and food and drug packaging industries. Direct interdependency between the gas barrier and the mechanical properties was not established enabling independent tailoring of these properties for mechanically rigid and impermeable thin film coatings.

## 1. Introduction

The growing performance and use of organic devices has required the development of efficient and ultra-thin gas barrier layers with adequate mechanical properties. Atomic layer deposition (ALD), a vapor phase deposition technique, is particularly suited for the preparation of ultrathin films of inorganic materials with sub-nanometer thickness control [1]. It is based on sequential pulses of precursors and co-reactants within the reactor chamber, enabling for self-limiting chemical reactions to take place at the substrate surface. The exceptional thickness control, but also the excellent uniformity and high conformality allowed by ALD, enabled this route to emerge as a key technology for the deposition of thin films for numerous applications, from microelectronics [2] to photovoltaics [3], and from biosensing [4] to membranes [5]. The key benefits of thin films prepared by ALD allow them to be used for gas barrier applications as well. Indeed, ALD films are very attractive candidates as gas barrier layers due to the conformal nature of the films on non-planar surface topographies [6]. Furthermore, the temperature window of many ALD processes allows the deposition of pinhole-free uniform thin films on flexible organic polymeric substrates. Groner et al. [7] was the first to demonstrate that 33 nm thick Al_2_O_3_ coatings on poly(ethylene-terephthalate) (PET) reduced the gas permeability for CO_2_. Water vapor transmission rates < 10^−5^ g H_2_O m^−2^ day, which are needed for packaging of organic light emitting diodes, were demonstrated by several groups for different film materials on polyethylenenaphtalate (PEN), for instance, Carcia et al. for a 25-nm thick Al_2_O_3_ film [8], Chou et al. for about 180 nm thick Hf doped ZnO films [9], and Behrendt et al. with 200-nm thick SnO_2_ ALD films [10]. For nanolaminate ALD films, similar gas barrier properties were shown for 40 nm thick Al_2_O_3_/ZrO_2_ [11], while 10^−4^ g H_2_O m^−2^ day was obtained for a 50 nm thick amorphous Al_2_O_3_/TiO_2_ alloy [12]. ZnO is a material presenting antibacterial activity and low toxicity, two excellent advantages for food and drug packaging [13]. In addition, this material is relatively easy to prepare by ALD on flexible substrates and, thus, also possesses this upscaling possibility benefit (using roll to roll processing, for example) [14]. As the presence of oxygen inside packaging is associated with the deterioration of the drug/food, reducing the quality of the product, good barrier materials against the permeation of oxygen are needed.

Mechanical studies on ALD films mostly concentrated on elastic modulus, hardness measurement, or residual stress and can be classified into work on single compound film materials, mainly oxides and nitrides, prepared on stiff substrates, like Si or glass [15,16,17,18,19,20,21,22,23,24,25,26] and Al_2_O_3_ [27]. The mechanical properties of nanolaminates films have also been investigated on stiff substrates [19,28,29,30,31,32,33,34]. Bull et al. [21] pointed out that reliable modulus and hardness data on modulus mismatched ceramic films on polymers can be obtained only if a suitable modeling approach is adopted. Due to the increasing use of flexible organic devices, there is a need to understand the mechanical properties of ALD films on such substrates. The reported work on flexible polymer substrates focused so far on fracture mechanics of single compound ALD films: Jen et al. [35] reported the critical fracture strains of Al_2_O_3_ ALD films for both tensile and compressive strains. The results show that the critical tensile strain is higher for thinner thicknesses of the Al_2_O_3_ ALD film on heat-stabilized polyethylene naphthalate (HSPEN) substrates. A low critical tensile strain of 0.52% was measured for a film thickness of 80 nm while it increased to 2.4% at a film thickness of 5 nm. The fracture toughness for tensile cracking, *K_IC_*, of the Al_2_O_3_ ALD film was determined to be *K_IC_* = 2.30 MPa m^1/2^. Miller et al. [18] shows measurements of crack density versus applied tensile strain coupled with a fracture mechanics model of alumina and alucone films, as well as a single-layer alucone/Al_2_O_3_ sandwich deposited on polyethylene naphthalate substrates. We report their values in the results section for comparison. Latella et al. [36] studied toughness and adhesion of ALD alumina films on polycarbonate substrates. The strength and toughness of the alumina film were determined to be 140 MPa and 0.23 MPa m^1/2^, respectively. Adhesion of the alumina films was improved for substrates pre-treated with water plasma. Ding et al. [20] reports an adhesion energy of an ALD alumina film on polyimide of 0.12 J/m^2^. TiO_2_ ALD films on polycarbonate substrates showed an enhanced interface energy of 26 J/m^2^ and interface toughness of about 2MPa m^1/2^ with a water plasma pre-treatment [37].

Fracture mechanics studies on multi-nanolaminate films on polymer substrates were not yet reported to our knowledge, except for a single alucone layer sandwiched by ALD Al_2_O_3_ films [18] and alucone Al_2_O_3_ nanolaminates [38].

In this work, we investigate Al_2_O_3_/ZnO nanolaminates on PET to evaluate their potential as barrier layers. First, various nanolaminates presenting thicknesses close to 250 nm were grown and carefully characterized in terms of their microstructure. Next, the mechanical properties, namely, fracture strain, critical bending radius, interfacial shear stress, and crack behavior as a function of their bilayer thickness were assessed and discussed. Finally, the gas permeability of the different nanolaminates has also been assessed for O_2_ gas.

## 2. Materials and Methods

### 2.1. ALD of Al_2_O_3_/ZnO Nanolaminate Films

Al_2_O_3_/ZnO nanolaminate films of various bilayer thicknesses were deposited in a custom-built stationary ALD system, using trimethylaluminium (TMA) and diethyl zinc (DEZ) as precursors for Al_2_O_3_ and ZnO, respectively. Water was used as the co-reactant for both processes, and all the depositions were achieved at 90 °C. The samples were held at 90 °C for 30 min before deposition. The deposition parameters were: precursor pulse/exposure/purge 0.1 s/30 s/40 s (TMA), 0.2 s/30 s/40 s (DEZ), and 2 s/30 s/40 s (H_2_O). The precursor pulse and the purge step were carried out with 25 sccm and 100 sccm argon flows, respectively, as a gas vector. For the nanolaminate coatings, the Al_2_O_3_ layer was deposited first in all cases.

The Al_2_O_3_/ZnO nanolaminate films were deposited simultaneously on Si wafers (for structural and chemical characterization) and on 175 μm thick PET (biaxially oriented polyethylene terephthalate, Goodfellow Cambridge Ltd.) strips with a gauge section of 3 × 35 mm^2^ (for the study of the mechanical properties). Both the Si wafers and PET strips were cleaned before ALD, by introducing them successively in water and ethanol in an ultrasonic bath for 3 min.

Several Al_2_O_3_/ZnO nanolaminates were grown with varying bilayer numbers and bilayer thicknesses: 50 × 4.8 nm, 10 × 25 nm, 2 × 130 nm while keeping the Al_2_O_3_ to ZnO ALD cycle ratio = 1. The total number of ALD cycles was fixed at 1000 cycles for all the depositions to achieve a total film thickness close to 250 nm for all coatings. Single ZnO and Al_2_O_3_ coatings were also deposited by using 1000 cycles and characterized as reference samples. The total film thicknesses of the Al_2_O_3_/ZnO nanolaminates were measured from SEM cross-sections and the mean bilayer thicknesses were calculated from the number of ALD supercycles (equal to the number of bilayers).

### 2.2. Microstructural Characterization

The Al_2_O_3_/ZnO nanolaminate films deposited simultaneously on silicon substrates were characterized using grazing incident X-ray diffraction (GIXRD, Bruker AXS, D8 Discover, Billerica, MA, USA) in order to study the structural evolution of the ZnO grain size when the bilayer thicknesses varied. GIXRD (Bruker AXS, D8 Discover, Billerica, MA, USA) measurements were achieved by using CuK_α_ (λ = 1.5418 Å) radiation with the excitation voltage and the current set at 40 kV and 40 mA, respectively. The angle of incidence for GIXRD was kept constant at 0.5°; the step size and scan ranges used were 0.02° and 25° to 50° degree, respectively. The thicknesses of films deposited on PET substrates were characterized using scanning electron microscopy (SEM, Hitachi S-4800, Tokyo, Japan).

Additionally, focused ion beam microscopy (FIB: Helios 600, FEI, Hillsboro, OR, USA) was used to section cracks and buckles in the coatings in order to investigate crack morphologies and interface failure modes. Pt-C material was locally deposited by Ga-FIB to protect the surface during cross-section and lamella milling. FIB was also used to prepare a transmission electron microscopy (TEM) lamella from the 50 × 4.8 nm bilayer sample. Scanning TEM (STEM, ThermoFisher Scientific Talos 200X, Waltham, MA, USA) was performed on the 50 × 4.8 nm sample at 200 kV to investigate the bilayer repeat and to analyze the modified polymer below the multilayer by electron dispersive X-ray spectroscopy (EDS).

The total film thicknesses of the Al_2_O_3_/ZnO nanolaminates and the single Al_2_O_3_ and ZnO films were measured on various fragments which delaminated from the PET substrate as a result of the tensile strain experiment. The determined thicknesses agreed within error to local FIB cuts we performed for microstructural and failure analysis. The bilayer thicknesses (one stack of Al_2_O_3_ and ZnO) were calculated by dividing the total film thickness by the number of bilayers.

### 2.3. Mechanical Tensile Tests

The Al_2_O_3_/ZnO nanolaminates deposited on PET substrates were tested using an MTI 8000-0010 tensile stage (MTI Instruments, Inc., Albany, NY, USA) with a fixed strain rate of 1.7 × 10^−4^ s^−1^. The surface of the sample was monitored with a digital optical microscope (Keyence 500F, Osaka, Japan) during the experiments. A series of images were recorded at strain intervals of 0.015%, and strain was determined using digital image tracking from one to three pairs of points on the sample surface. At least six stripes of each sample were strained and analyzed for obtaining the data for crack density as a function of applied strain.

The tensile testing results were analyzed using a two-parameter Weibull distribution for coating strength. This allowed the determination of cohesive strength of the films from the evolution of fragmentation that yielded the Weibull distribution shape parameters. The procedure is explained in detail elsewhere [39].

### 2.4. Gas Permeability Measurement

The permeability of the films has been measured on PET for O_2_ (99.999%) by using the constant-volume and variable-pressure technique in a permeability apparatus at 25 °C; following the standard ASTM D 1434-82 (procedure V). The apparatus consists of a two-compartment stainless steel permeation cell separated by the ALD-coated PET membrane (surface 1.59 × 10^−3^ m^2^ and radius 2.25 × 10^−2^ m) and sealed with silicone O-rings. The permeability was obtained by measuring the pressure increase in the downstream compartment (with a constant volume of 5.25 × 10^−5^ m^3^) and using different MKS Baratron pressure transducers (ranging from 0.0 to 1 × 10^5^ Pa) for two or three samples per ALD film material. The ALD-coated PET membrane and downstream cell walls were outgassed in situ during 24 h at high vacuum using a turbomolecular pump (Leybold, Turbovac 50, 50 l s^−1^, Cologne, Germany). The permeability experiments were performed using 3 × 10^5^ Pa of upstream pressure and recording the pressure increase in the downstream compartment during 24 h. Curves obtained present a pseudo steady-state situation. For calculations of the permeability, the mathematical treatment for thin films based on Fick’s second law [40] was used.

## 3. Results

### 3.1. ALD Film Thickness and Microstructure

The total film thickness of the single oxides was measured to around 200 ± 19 nm and 210 ± 38 nm for Al_2_O_3_ and ZnO, respectively, with relatively large point-to-point variations. For Al_2_O_3_ we attribute this to the sub-surface growth, as discussed later in this section, while for ZnO the large variations could be attributed to the locally varying nanocrystalline growth. For the single Al_2_O_3_ film the growth per cycle (GPC) of 2 ± 0.2 Å at 90 °C on PET is large compared to reported values of 1.3 Å to 1.4 Å for 177 °C [41] and 60 °C [29], respectively, on Si samples. The average ZnO the GPC of 2.1 ± 0.4 Å is only slightly larger than reported in [29,41] but the 20% variation in our ZnO GPC is considerably larger. We attribute the larger Al_2_O_3_ GPC and the large ZnO GPC variations to the PET acting very differently than Si for ALD nucleation. We observed the total thickness of the nanolaminates to be about 20 to 30% larger, see Appendix A. This may be related to an increase in GPC for the Al_2_O_3_ layers with increasing number of ZnO layers which can be up to about 36% inferred from data by [29] at 60 °C. On the other hand, the GPCs of ZnO in nanolaminates were found to reduce by 15% [29]. An effective nanolaminate GPC increase resulting from the both GPCs of Al_2_O_3_ and ZnO would thus be of about 20% which is in good agreement with the total thickness increase we observed. Similar trends for the change of GPCs of Al_2_O_3_ and ZnO at higher temperatures were also found by [41,42] and attributed (like in our case) to the higher surface area of the rough crystalline ZnO interfaces.

The grain size has been determined using XRD data and the Scherrer equation. Figure 1 shows the GIXRD spectra of the Al_2_O_3_/ZnO nanolaminate films for varying bilayer thicknesses. The nanocrystalline ZnO single layer film shows the ZnO peaks corresponding to ‹100›, ‹002›, ‹101›, and ‹102› orientation. The preferred growth orientation for the single ZnO film is ‹101› matching the results reported for ZnO of 200 nm thickness in [43]. The Al_2_O_3_ single film is amorphous with no peaks appearing.

For the smallest bilayer thickness of 4.8 nm, there was a broad ‹002› ZnO peak observed as preferential orientation. A previous transmission electron microscopy (TEM) study reported a transition to the amorphous ZnO state at ≤ 2.5 nm bilayer thickness [29]. The intensity of the ‹002› peak increased for the 25 nm bilayer thickness. For the 130 nm bilayer thickness the ‹100› peak became the preferred orientation. The suppression of ZnO crystallinity by multilayering Al_2_O_3_ and ZnO is a well know phenomenon, it was published already at one of the early works of ALD ZnO [44] and has been then repeatedly reproduced since [29,42,45,46,47]. The ZnO grain size *D* in the nanolaminate films was estimated from the Debye-Scherer equation with: D = 0.9λ⁄(Bcosθ), where λ, B, and θ are the X-ray wavelength, ‹002› peak full width at half maximum, and diffraction angle, respectively. The grain sizes for the bilayer thicknesses 4.8 nm and 25 nm were found to be 2 nm and 11 nm, respectively, which is smaller than the film thickness of the ZnO layers sandwiched between the amorphous Al_2_O_3_ layers. This can be understood taking into account that the grain growth of ZnO in nanolaminates is interrupted by the amorphous Al_2_O_3_ layers. The 130 nm bilayer thickness had a ZnO grain size of 19 nm which is only about one third of the ZnO film thickness. Moreover, the grain size of 14 nm calculated for the single ZnO film was only 1/15 of its total film thickness. We attribute this to the above stated different preferential growth orientations of ZnO in nanolaminates compared to the single ZnO film.

Figure 2 shows STEM cross-sections and SEM images of FIB cut cross-sections of the samples summarizing the key finding of their structural analysis. Figure 2a,b show a STEM of the nanolaminate structure of the 50 × 4.8 nm sample. The nanolaminate periods can be easily distinguished, and all but one of the intended 50 layers can be identified. Accordingly, 20 cycles have not been enough not properly nucleate uniform ALD growth, additionally the few following bilayers are not regular indicating that steady growth is obtained only after some 60 ALD cycles. Additionally, Figure 2b,d show a thick sub-surface growth that has been formed by the diffusion of TMA into the PET. No such sub-surface growth could be observed for ZnO films shown in Figure 2c and Figure 5a. Additionally, EDS line scan shown in Appendix A for a nanolaminate sample shows no or little Zn within the sub-surface growth. Subsurface diffusion of ALD precursors into polymers and other soft substrates is a relatively well known phenomenon [48,49,50,51,52]. For instance Sun et al. reported very similar looking sub-surface growth deposited on polyamide-6, and discovered that thinner subsurface growth had higher fracture strain [51]. However, while work has been done on ALD on PET substrates from the point of diffusion barriers [7,53,54,55,56,57], flexible electronics [58,59,60,61], mechanical properties [21,62] and surface modification of textiles [63], the subsurface diffusion of the precursors seems not to have been much quantified. From Figure 2 and EDS line scans across the same nanolaminate (Appendix A) we observe significant TMA and H_2_O diffusion down to a 900 nm depth being possible during the 30 s exposure to TMA at 90 °C in our conditions, while DEZn does not diffuse into the PET within our approximately 10 nm observation resolution.

### 3.2. Mechanical Properties

Next, the mechanical properties of the nanolaminates have been investigated. The development of cracks and buckles can be seen optically, as shown exemplarily in Figure 3, which presents images of the 50 × 4.8 nm bilayer sample during tensile testing at three different levels of strain. The very straight and parallel cracks with their normal parallel to the tensile strain direction are typical of all the coatings tested—this type of cracking is typical of brittle fracture tested in uniaxial tension [64,65]. The buckling occurs due to compressive strains as a result of substrate contraction perpendicular to the tensile direction.

Tilt view SEM images in Figure 4 show a different behavior of ZnO and Al_2_O_3_ single films in terms of adhesion. The buckles of the ZnO were relatively small and cracked which points to good adhesion to the PET. In contrast, the Al_2_O_3_ film developed large delamination buckles upon strain, which is only possible for weak adhesion.

In Figure 5 it can also be seen that for all the samples the crack continues sub-interface, i.e., within the PET substrate. This is the main failure mechanism observed in the samples, and corresponds to the cracks shown in Figure 3c. We attribute this behavior to the mismatch of the mechanical properties of the thin films and the PET substrate, which allows the cracks initiated in the strong thin film to penetrate into the soft polymer substrate [66].

The sub-surface growth was observed to have major effect to the delamination of the films as it introduced new failure modes to the system as illustrated in Figure 6. From Figure 6 it is evident that failure of the sub-surface growth during tensile strain may lead to delamination or the failure of the thin film. Accordingly, the sub-surface growth likely explains the lower fracture strain and lower interfacial shear strain of Al_2_O_3_ and the nanolaminates in comparison to ZnO. Nevertheless, this does not dictate that sub-surface growths are always detrimental for the mechanical properties of the film. In this case the sub-surface growth was very thick, which is expected to lead to low fracture onset strain. Further studies would be needed to verify if thin sub-surface growth would have a detrimental effect.

The evolution of the crack density as a function of strain is presented in Figure 7. The strain was measured on the video sequences by image tracking. Crack formation starts at a certain fracture (or crack) onset strain and increases with increasing strain. The crack density saturates at *CD_sat_*, as predicted from shear lag theory [67], for all the tested coatings at 8–9% applied strain. Table 1 and Figure 8 summarize fracture strains and saturated crack densities for all samples. The measurement results for each individual test piece can be examined in Appendix A. The highest fracture onset strain and crack density at saturation were observed for the single ZnO. It is notable that the crack density at saturation varies between the coatings by a factor of 3.5 from 37 to 142 mm^−1^ despite the coating thickness only varying by around ±30% between the samples. According to the theory [35], the saturation crack density for single thin films varies with their total film thickness as *h_f_**~** CD_sat_*^−0.5^ which suggests only 14% changes of the saturated crack density instead of the large observed differences in our samples. This is due to the better adhesion of ZnO with the PET substrate, which, in turn, is explained by the absence of the sub-surface growth.

We have also evaluated in Table 1 the critical bending radius *R* for the samples from the fracture onset strains *ε* by *R* = (*h_f_* + *h_s_*)/(2*ε*), where *h_f_* and *h_s_* represent the thicknesses of the film and substrate, respectively. Single ZnO showed the lowest bending radius of about 11 mm while the sample with 130 nm bilayer thickness had the highest radii equivalent to about 18 mm. These values allow the evaluation of the material for flexible applications as curvatures below these radii will initiate cracking of the coatings. 

The crack density versus strain graphs shown exemplarily in Figure 7b can be used to calculate the interfacial shear strength (or stress) *τ_IFSS_* [67,68]: *τ_IFSS_* = *1.337**ε_c_**E_f_**h_f_**CD_sat_*(1)
where *ε_c_* is the cohesive strain of the film obtained from a Weibull analysis of the crack densities close to fracture onset strain [39,68], *CD_sat_* is the saturation crack density, *E_f_* is the Young’s modulus, and *h_f_* is the film thickness. For the details of the Weibull analysis we refer the interested reader to Leterrier et al. [68] and of interfacial shear strength determination to [64,65,69]. Interfacial shear stress values *τ_IFSS_* were calculated for each of the coatings, taking Young’s moduli from on a very similar nanolaminate sample set published previously by us [29], and are reported in Table 1.

Cohesive film strength shown in Table 1 is a parameter that takes into account the kinetics of the crack evolution of the films, i.e., how well the films resist failure after the films have initially cracked. Cohesive strength of the nanolaminate samples seems to improve with smaller nanolaminate period, which is also noticeable in Figure 7b as lower slope of the crack density—strain curve. Especially the 50 × 4.8 nm nanolaminate shows higher cohesive strength than either of its constituents.

The graphical representation of the values of Table 1 is shown in Figure 8. Figure 8a illustrates that ZnO clearly needs the highest strain to be fractured. Moreover, two of the nanolaminates perform better than pure Al_2_O_3_ or the thickest nanolaminate with 100 nm bilayer thickness. The samples with 25 or 4.8 nm bilayer thicknesses take values between those of ZnO and Al_2_O_3_ and, thus, follow rather the rule of mixtures than weakest link prediction. Interestingly, the thickest nanolaminate with 100 nm bilayer thickness seems to take the fracture strain value of its weakest constituent, Al_2_O_3_. George et al. observed critical fracture onset strain and saturated crack density for different thicknesses of Al_2_O_3_ films on PEN to vary with (*h_f_*)^−0.5^**** and found values for 5, 25, and 80 nm thick single Al_2_O_3_ films as 2.4%/880 mm^−1^, 1.2%/370 mm^−1^, and 0.52%/230 mm^−1^, respectively. Scaling to 200 nm Al_2_O_3_ films one would expect 0.32%/145 mm^−1^ for fracture onset strain and saturated crack density. However, for our 200 nm thick single Al_2_O_3_ films on PET we found 0.49%/39 mm^−1^ in comparison which points to a weaker adhesion of Al_2_O_3_ films to PET than PEN.

From Figure 8b, we observed that interfacial shear stresses for ZnO were roughly double to triple than for the other samples, which is likely explained by the nanolaminates’ first layer to PET being always Al_2_O_3_ in our ALD process. This difference is also manifested in the SEM images taken after the tensile experiments; ZnO remains well adhered to the substrate, while Al_2_O_3_ has mostly buckled off; see Figure 4. Accordingly, for the nanolaminates the interfacial shear stresses took values close to Al_2_O_3_, i.e., the value was determined by material that lies at the interface. Compared to values of Miller et al. who obtained the interfacial shear stress for 5 nm, 25 nm, and 125 nm single Al_2_O_3_ on PEN films as 25.0 ± 1.8 MPa, 32.3 ± 4.3 MPa, and 61.1 ± 8.3 MPa, respectively, the value of 15 ± 2 MPa for 200 nm thick single Al_2_O_3_ on PET is rather small. However, the above Al_2_O_3_ on PEN literature values compare to our single ZnO film on PET samples. Thus, rather interestingly, under the deposition conditions used in this study, ZnO has far better adhesion to PET than Al_2_O_3_, and indicates a potential use for ZnO as an adhesive layer for ALD films to be deposited on PET. The same trend has been observed in the graph between saturated crack density vs. bilayer thickness; see Figure 8c in which ZnO has clearly the highest value, while differences among the other samples are subtle. The sub-surface growth is likely explaining the low interfacial shear stress values for Al_2_O_3_ and the nanolaminates. Cracks may initiate at the sub-surface growth leading to lower saturation crack density. Additionally, the sub-surface growth brings along new failure mechanisms as illustrated in Figure 6. Calculating the interfacial shear stress values by including the thickness of the sub-surface growth would lead to roughly 2–3 times larger values and would come close to the values for ZnO. 

### 3.3. Gas Permeability Testing

In Figure 8d, the gas permeability of the tested coatings on PET from Table 1 is plotted on a logarithmic scale; this highlights the improvement of the nanolaminate coating properties compared to the single coatings. The single compound Al_2_O_3_ and ZnO films already reduce O_2_ permeability of PET by a factor of 500 to 1000. The nanolaminate coatings on PET offer an O_2_ gas barrier approximately ten times more impermeable than the two single oxides on PET which amounts to an improvement of four orders of magnitude than PET alone. It seems that the improved barrier properties of the nanolaminates cannot be related to the sub-surface growth material as this would have improved already the single compound Al_2_O_3_ film. Between the Al_2_O_3_/ZnO nanolaminate coatings there is no significant variation in permeability measured. The measured values for O_2_ permeation of our ALD films on PET in the range of 2.5 × 10^−2^ to 3.1 × 10^−1^ cm^3^ m^−2^ day^−1^ are consistent with those published for ALD fabricated gas diffusion barriers; for 100 nm and 130 nm thick Al_2_O_3_/ZrO_2_ nanolaminates 2.1 × 10^−2^ and 1.6 × 10^−2^ cm^3^ m^−2^ day^−1^ has been reported, respectively [70]. However, the results are not directly comparable as in [71] ALD films were directly deposited and tested on Ca substrates which entails completely different film nucleation and growth when compared to polymer carriers. In view of the large thickness variations and the sub-surface growth we observed for our barrier films on PET and discussed in Section 3.1, our permeation values characterize the entire materials system with its specific growth and nucleation features involved and seem closer to practical use in packaging applications. For measurements of barrier coatings on polymer substrates, the best single Al_2_O_3_ film permeation values are reported in the range of 0.1 to 1 cm^3^ m^−2^ day^−1^ (for PET the reported value being 1.8 cm^3^ m^−2^ day^−1^) [71,72,73,74], which we outperform by about one to two orders of magnitude using the Al_2_O_3_/ZnO nanolaminates. Values as low as 5 × 10^−3^ cm^3^ m^−2^ day^−1^ have been singularly reported for 10-nm thick films [53], however, using a Ca-corrosion test.

For Al_2_O_3_/ZrO_2_ nanolaminates Meyer et al. [11] proposed the advantages of nanolaminates for gas diffusion barriers as follows, (i) the suppression of crystallization in the alternating Al_2_O_3_/ZrO_2_ sublayers, (ii) the protection of Al_2_O_3_ from water corrosion by the ZrO_2_, and (iii) the formation of an aluminate phase at the Al_2_O_3_/ZrO_2_ sublayer interfaces. For the here-presented Al_2_O_3_/ZnO nanolaminates we can rule out these three factors for barrier improvement against O_2_. Our nanolaminates were still nanocrystalline ZnO, and the single films nanocrystalline ZnO and amorphous Al_2_O_3_ showed approximately the same O_2_ permeability. Additionally, we did not detect any ZnAl_2_O_4_ spinel formation. We assume grain boundaries in ZnO and the comparably lower density of Al_2_O_3_ ALD material containing about 15 at.% OH at 90 °C [7,75] compared to sapphire to be responsible as main diffusion mechanisms.

Although a thorough study of the mechanism behind the enhanced gas barrier behavior of the nanolaminate coatings is beyond the scope of this work, the authors would, however, like to put forward an explanation. The size of defects in the coatings is very important when considering gas permeability. Defects provide a fast channel for the diffusion of gas species through a membrane; this is also true of grain boundaries but the lack of variation in permeability between the nanolaminates, where the ZnO grain size does vary, suggests that this does not play an important role. The lack of variation in permeability between the nanolaminate coatings also suggests that the size of the defects within the nanolaminate (assumed to be equal to the bilayer thickness or the grain size) is also not the controlling factor. However, if we also assume that the distribution of defects within the nanolaminate layers is random then one can imagine that the likelihood of a permeability increasing defect running straight through the entire thickness of the nanolaminate is rather low as the amorphous Al_2_O_3_ layers are disrupting it. It could be this reduction in through-thickness diffusion defects that leads to the observed reduction in gas permeability of the nanolaminate coatings compared to the single ceramic coatings. Of interest is here that such a reduction of through-thickness diffusion defects should also manifest in increased fracture strain as set forth by the seminal work of Griffith [76]. However, we did not observe such a relation which may point to a different nature of defects responsible for diffusion and fracture.

## 4. Discussion

Al_2_O_3_/ZnO nanolaminates with thicknesses of around 250 nm were deposited on PET foils by atomic layer deposition at 90 °C using TMA, DEZ, and H_2_O. Bilayer thicknesses were varied from 4.8 nm to 130 nm. A non-linear increase of the nanocrystal size for ZnO was found with increasing bilayer thickness which was related to a change in preferential growth orientation. Al_2_O_3_ was always deposited amorphously. Uniaxial tensile testing of the coatings revealed that all films behaved in the same basic manner: long parallel cracks form in the coatings and the density of these cracks per unit length increases until a saturation point is reached at 8–9% applied strain. Single ZnO films were found to have superior fracture resistance and adhesive properties than the single Al_2_O_3_ and their nanolaminates. Nanolaminates with thick bilayers were found to have similar fracture properties than their weakest Al_2_O_3_ constituent, while ones with thinner nanolaminate bilayers took values close to the average values of the constituents. The interfacial properties of the nanolaminates were determined by the material that resided at the interface. The superior mechanical performance of ZnO was linked with absence of formation of sub-surface growth. The small crack onset strain, considerable delamination, and the small saturated crack density of Al_2_O_3_ film and the nanolaminates could be explained by sub-surface growth and related phenomena. Al_2_O_3_ and the nanolaminates were found to have 500–900 nm thick sub-surface growth. The critical bending radii ranged from about 11 mm for 210 nm single ZnO to about 18 mm for 200 nm single Al_2_O_3_ and the 2 × 130 nm thick bilayer nanolaminate.

Oxygen gas permeability measurements of the ALD coatings on PET revealed that the nanolaminates have good barrier properties with 9.4 × 10^−3^ O_2_ cm^3^ m^−2^ day^−1^. This is an order of magnitude less permeable to O_2_ than their constituting single oxides which themselves already rendered the 175 μm thick PET foil about 1000× less permeable. This property of the Al_2_O_3_/ZnO nanolaminate coatings highlights their potential for use as gas barrier. The mechanical and gas barrier properties do not seem to correlate via a through-thickness defect model which proposes different mechanisms of mechanical failure and diffusion. An explanation based on non-overlapping defects was put forward for the understanding of the barrier enhancement.

We have analyzed the mechanical fracture, crystallinity and O_2_ permeability of the materials. However, no clear interdependencies of these material properties were found. Both O_2_ permeability and ZnO crystallinity were reduced by nanolaminating, but crystallinity does not seem to explain the reduction in the permeability. This suggests that all these three parameters may be tuned independently in this material system allowing the design of mechanically rigid and impermeable thin film coatings. 

The results presented in this work open new avenues for the applications of ALD films as gas barrier layers for organic electronics, but also for food or drug packaging industries. 

## Figures and Tables

**Figure 1 nanomaterials-09-00088-f001:**
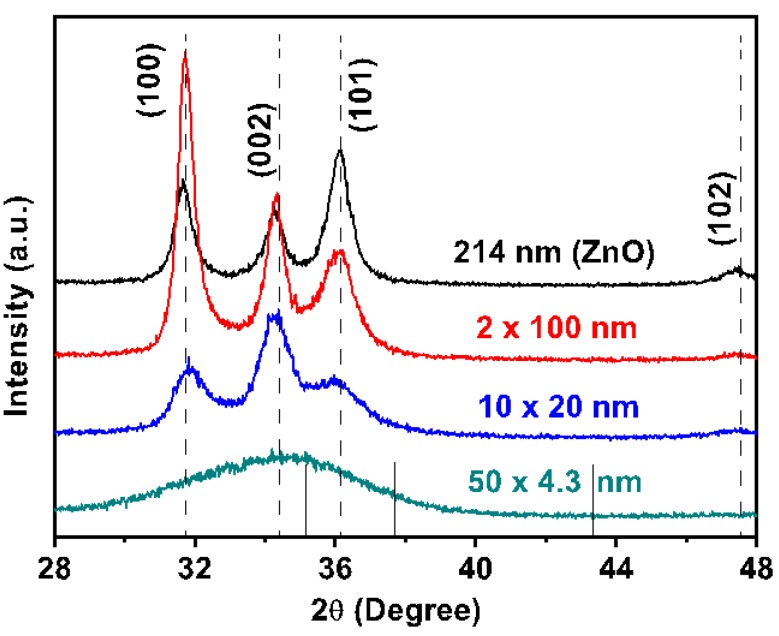
GIXRD spectra of ZnO peaks in Al_2_O_3_/ZnO nanolaminate films having varying bilayer thicknesses (indicated). The long dashed lines indicate bulk ZnO diffraction patterns (ICSD 67454) and the short solid lines indicate those for bulk alumina (ICSD 51687).

**Figure 2 nanomaterials-09-00088-f002:**
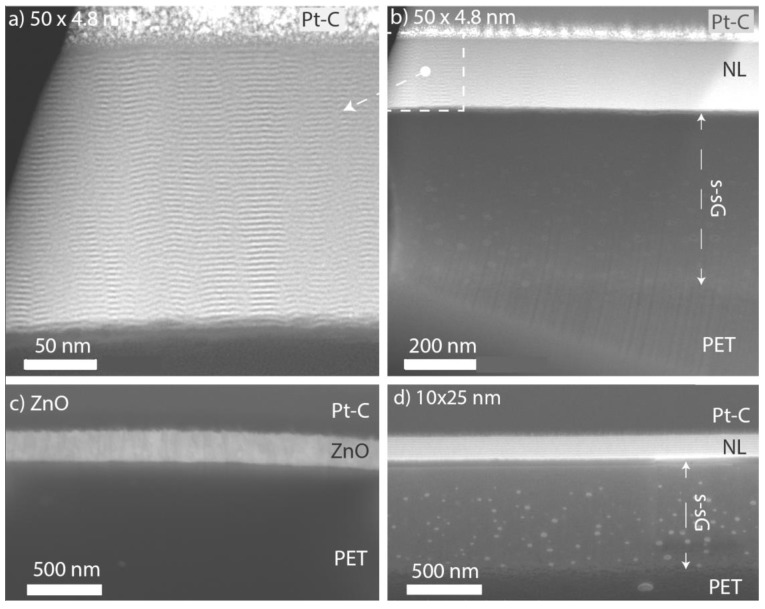
(**a**) and (**b**) STEM cross-sections of the 50 × 4.8 nm Al_2_O_3_/ZnO nanolaminate on PET sample (NL = nanolaminates, s-sG = sub-surface growth, Pt-C = local protection material deposited by FIB for FIB cut). SEM of FIB cut cross-sections of (**c**) the ZnO film and (**d**) the 10 × 25 nm nanolaminate (52° angle). Images were taken after tensile strain experiments. Note the sub-surface growth (s-sG) for ALD films starting with Al_2_O_3_ in (**b**,**d**).

**Figure 3 nanomaterials-09-00088-f003:**
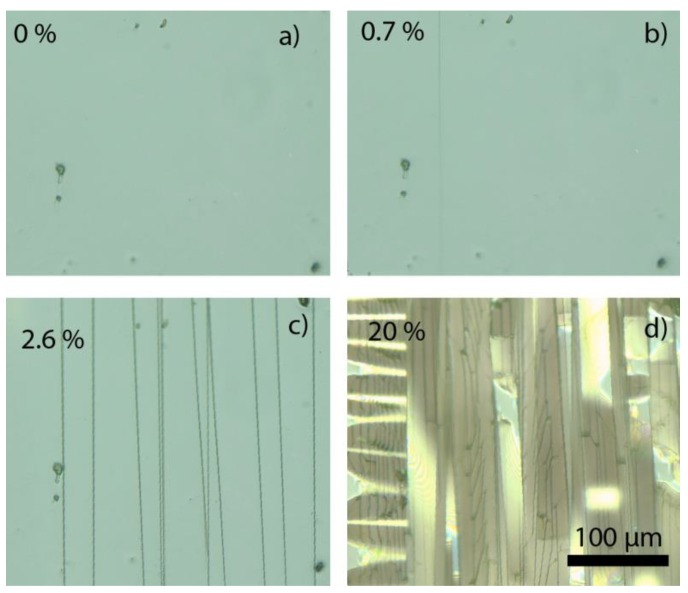
Optical microscope images of 240 nm thick Al_2_O_3_/ZnO nanolaminate with 4.8 nm bilayer thickness during tensile strain experiment (tensile strain is applied horizontally in these images). Note the increasing density of cracks from (**a**,**c**) and then the development of buckles and delamination of the film (bright shiny features) from the substrate in (**d**). Cracks in the sub-surface growth material can also be seen in (**d**); compare to Figure 6b.

**Figure 4 nanomaterials-09-00088-f004:**
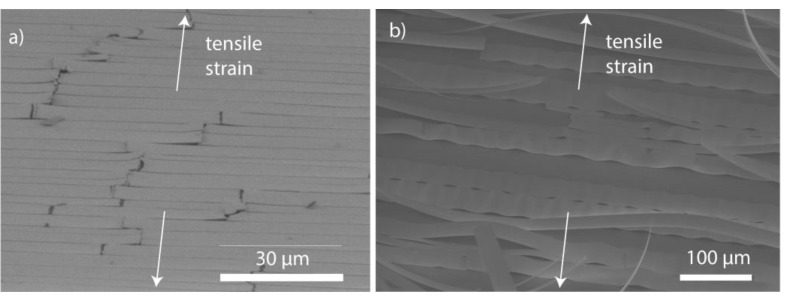
SEM overview of the cracked and buckled surfaces on the PET substrates after the tensile testing up to 12%. (**a**) ZnO sample (from 70° tilt angle) and (**b**) Al_2_O_3_ sample (from a 45° tilt angle).

**Figure 5 nanomaterials-09-00088-f005:**
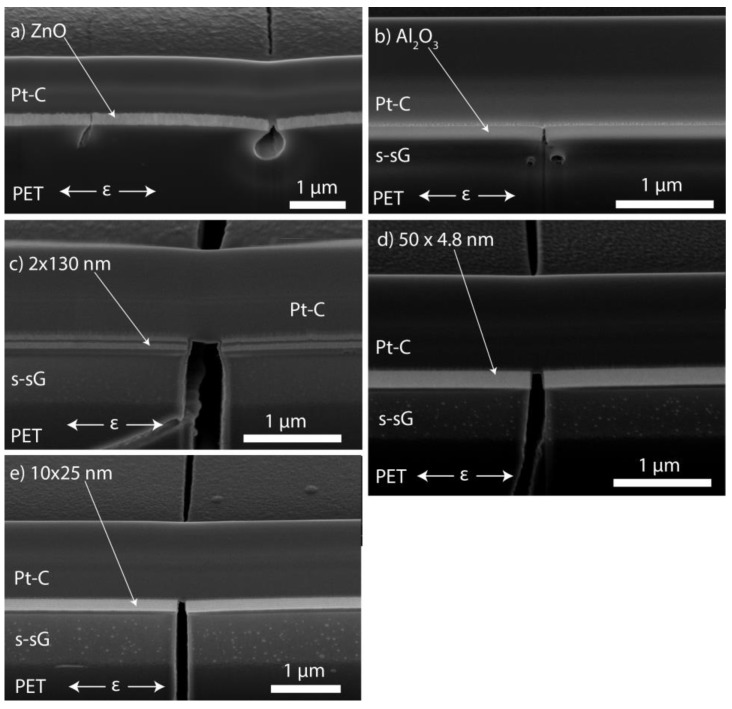
SEM images of FIB cut cross-sections (52° angle) for all the samples; (**a**) ZnO, (**b**) Al_2_O_3_, (**c**) 2 × 130 nm nanolaminate, (**d**) 50 × 4.8 nm nanolaminate, (**e**) 10 × 25 nm nanolaminate. (s-sG = sub-surface growth, Pt-C = local protection material deposited by FIB for FIB cut) showing through-thickness through substrate cracks during tensile strain.

**Figure 6 nanomaterials-09-00088-f006:**
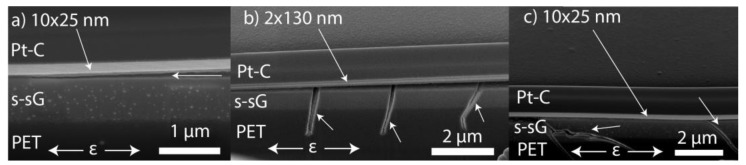
SEM images of FIB cut cross-sections (52° angle) of the indicated samples (s-sG = sub-surface growth, Pt-C = local protection material deposited by FIB for FIB cut) showing additional failure modes (marked by arrows) brought by the subsurface growth. (**a**) Failure of the sub-surface growth material that delaminates the film (**b**) crack failure of the sub-surface growth material that propagates to the PET substrate, (**c**) failure of the sub-surface growth material that partially delaminates the film.

**Figure 7 nanomaterials-09-00088-f007:**
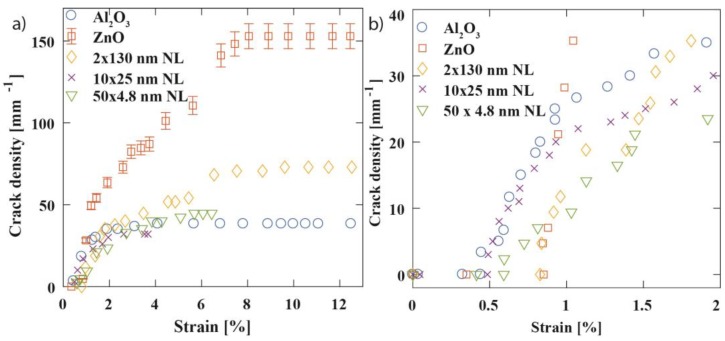
Crack density evolution during tensile tests for a selection of test samples. (**a**) Full strain range from zero to 10%. (**b**) Zoom into the region of fracture onset strain.

**Figure 8 nanomaterials-09-00088-f008:**
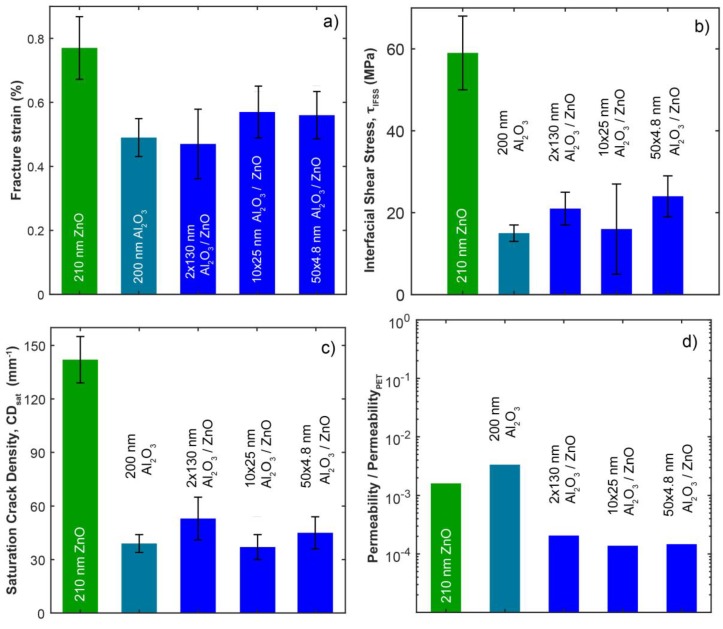
Graphs of (**a**) fracture onset strain, (**b**) interfacial shear stress, and (**c**) saturation crack density versus bilayer thickness for the roughly 250 nm thick nanolaminate coatings and the single (single compound) films ZnO and Al_2_O_3_ on PET, (**d**) Permeability of O_2_ of the single ZnO and Al_2_O_3_ and the 130, 25, and 4.8 nm bilayer thickness nanolaminate coatings on PET compared to the permeability of the 175 µm thick PET. All the nanolaminates were about 250 nm thick in total (see Appendix A). Note the logarithmic scale for (**d**).

**Table 1 nanomaterials-09-00088-t001:** Uniaxial tensile testing data: crack onset strain *ε_f_*, saturated crack density *CD_sat_*, and critical bending radius *R*, calculated interfacial shear stress *τ_IFSS_* (Equation (1)), measured thicknesses, estimated thicknesses of sub-surface growths, O_2_ permeability, grain size, and Young’s modulus *E_f_* (from [29]) for the investigated coatings. A denotes amorphous by XRD, and the O_2_ permeability of the 175 μm thick PET substrate was 6.35 × 10^−4^ Barrer. The permeability values relate to the ceramic coatings, the PET contribution was taken off (1 Barrer = 10^−10^ cm^3^(cm·s·cm_Hg_)^−1^).

Sample	Grain Size (nm)	Fracture Strain, ε_f_ (%)	Saturation Crack Density, *CD_sat_* (mm^−1^)	Cohesive Strength (MPa)	Interfacial Shear Stress, *τ*_IFSS_ (MPa)	Layer Thickness, h (nm)	Sub- surface Growth, Thickness (nm)	Young’s Modulus, E_f_ (GPa)	Bend Radii (mm)	O_2_ Permeability (Barrer)	O_2_ Permeation Rate (cm^3^ m^−2^ day^−1^)
Number × Thickness of Bilayers											
ZnO	14	0.77 ± 0.1	142 ± 13	1460 ± 280	59 ± 9	210	-	145	11.4	1.02E−06	1.10E−01
Al_2_O_3_	A	0.49 ± 0.06	39 ± 5	1460 ± 330	15 ± 2	200	300	164	17.9	2.13E−06	2.30E−01
2 × 130 nm	19	0.47 ± 0.11	53 ± 12	1100 ± 376	21 ± 4	260	670–820	152	18.6	1.31E−07	1.40E−02
10 × 25 nm	11	0.57 ± 0.08	37 ± 7	1450 ± 850	16 ± 11	250	880–1200	146	15.6	8.74E−08	9.40E−03
50 × 4.8 nm	4	0.56 ± 0.07	45 ± 9	1670 ± 450	24 ± 5	240	630	141	15.4	9.31E−08	1.00E−02

* Interfacial shear stress was calculated using the film thicknesses without considering the thickness of the sub-surface growth.

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
