# Peer review of "Fracture Mechanics and Oxygen Gas Barrier Properties of Al2O3/ZnO Nanolaminates on PET Deposited by Atomic Layer Deposition"

_nanomaterials, 2019, doi:10.3390/nano9010088_

Reviewer 1 Report

The article reports on the mechanical properties and gas barries properties of Al2O3/ZnO nanolaminates. A complex mechanical behavior of the composite material was tackeld mostly from the angle of tensile tests and the analysis of those results. The complementary microstructural characterization is however weak and could have been far more detailed. Without that, the discussion of the results is based essentially on speculations, e.g. the formtion and role of defects in the film. As it is, the authors cannot really explain their data i.e. why one sample outperformed others in the gas permeability tests. While very interesting work worthy of publication, further more detailed study is highly recomended. Otherwise the article feels more like a sumup of a large volume of numerical data.

The authors should also explain why they chose the materials such as ZnO and why Oxygen for gas barrier tests. Have nanolaminated with ZnO as the first layer been tested as well?  

Author Response

A complex mechanical behavior of the composite material was tackeld mostly from the angle of tensile tests and the analysis of those results. The complementary microstructural characterization is however weak and could have been far more detailed. Without that, the discussion of the results is based essentially on speculations, e.g. the formation and role of defects in the film.

We added comprehensive FIB cross section analysis of all coatings on PET on crack and delamination regions as well as specific TEM observation (structural and subsurface composition) to address the reviewer’s comment on weak microstructural characterization (text and figures 2,5 and6 page 6-9). As mentioned in the manuscript, from these observations we can additionally state:

• The Al 2 O 3 film and the nanolaminate samples all form a thick ~500 nm subsurface growth under
the film itself. This layer is visible in both FIB and TEM. The layer is also fracturing during straining.
STEM-EDS reveals the layer to be of Al-C-O chemistry with no resolvable crystalline ordering.
• No such layer was observed for the ZnO film.
• In the nanolaminate films, it was observed that the multilayer/subsurface growth interface can
delaminate, revealing the cracked Al-C-O sublayer beneath. Thus, failure of the subsurface layer is able to cause the mechanical failure of the whole film stack. These new failure modes brought by the
subsurface layer are proposed to explain largely the differences in the mechanical behavior of the
samples.

As it is, the authors cannot really explain their data i.e. why one sample outperformed others in the gas permeability tests. While very interesting work worthy of publication, further more detailed study is highly recommended. Otherwise the article feels more like a sum-up of a large volume of numerical data.

With the above added observations we can conclude more firmly on the mechanical behavior as being
influenced by the subsurface region. We can furthermore exclude a direct relation between (through-
thickness) defects causing mechanical failure and defects supporting gas diffusion through the film. Thus
we propose grain boundaries (ZnO) and low material density due to high OH content (Al2O3) to act as
diffusion mechanisms in the single compound films and argue that in nanolaminates the alternation of
nanocrystalline and amorphous material leads to disruption of diffusion paths which enhances the
barrier properties.

The authors should also explain why they chose the materials such as ZnO and why Oxygen for gas
barrier tests. Have nanolaminated with ZnO as the first layer been tested as well?

We thank the reviewer for this comment. Through this study, we have come to the conclusion that films with ZnO as the first layer would avoid subsurface growth and further work is planned using this type of structures. We chose the materials such as ZnO and Oxygen for gas barrier tests mainly because we wanted to do a study aimed to real application such as drug or food packaging. We added the following text and appropriate reference in the Introduction in order to explain this more clearly (page 2):
“ZnO is a material presenting antibacterial activity and a low toxicity, two excellent advantages for food
and drug packaging. In addition, this material is relatively easy to prepare by ALD on flexible substrates, and thus also possesses this upscaling possibility benefit (using roll to roll processing, for example). As the presence of oxygen inside packaging is associated with the deterioration of the drug/food, reducing the quality of the product, good barrier materials against the permeation of oxygen are needed.”
REF: T. Tynell and M. Karppinen, Atomic layer deposition of ZnO: a review. Semiconductor Science and Technology, 29, 4 (2014)

Reviewer 2 Report

This is a well-written manuscript on alumina/zinc oxide nanolaminates deposited by ALD. It includes some interesting data and it probably requires further experiments for other aspects of the presented work.

1) the observation that for increasing bilayer thickness, the ZnO nanocrystal increases has been reported as early as 10-15 years ago for similar systems (e.g., aluimina/hafnia)

2) Reference 52 includes data fro alumina with permeation rates of 1.2x10^(-1) [at 100 nm] and 0.33x10^(-1) [at 130 nm] under pretty much the same ALD conditions and identical precursors. This manuscript shows a permeation rate of 6.9x10^(-1), that is much higher value, for a much thicker film of 200 nm! These values suggest that the permeation rates may be +/- 1 to 2 orders of magnitude, which, in turn, suggests that no significant gain, if any, is attained with the nanolaminates presented in this manuscript.

3) The data in Figure 5c for 2x130 nm samples seem to be in conflict with the same data included in figure 4a (i.e., saturation crack density of 50 vs. 65, respectively).

4) the inclusion of standards for bulk alumina and zinc oxide in figure 1 would be helpful

5) In figure 4a, the alumina and zinc oxide films in series can explain the results presented, when crystalline ZnO starts forming. (For example: 1/40 + 1/150 = 2/65)

6) Table S1 raises a few questions:

(a) the first two rows suggest .21 nm/cycle (210/1000) for zinc oxide and .20 nm/cycle (200/1000) for alumina.  The third row suggests .26 nm/cycle (250/[25+25)*10]) for the nanolaminate, which is 25-30% higher than the individual oxides; why and what does that mean for the ALD in question?

(b) In the fourth row, the data suggests .50 nm/cycle (250/[(25+25)*10]).  Why is this almost 150% higher than the individual ALD rates?

(c) What is the uncertainty of all data included in Table S1?

6) Finally, the above issues put in question the plausible mechanism presented in the later sections of the manuscript.  

Author Response

1)      the observation that for increasing bilayer thickness, the ZnO nanocrystal increases has been reported as early as 10-15 years ago for similar systems (e.g., aluimina/hafnia)

We agree with the reviewer comment. Indeed, the ZnO nanocrystal increase has already been reported in the literature. While we noticed a ZnO crystal size increase with the bilayer thickness, the main contribution of the paper was to study the relations between fracture mechanics and permeability of nanolaminates. The crystallinity of the nanolaminates was analyzed in order to access the potential correlation with it to the other studied parameters; however, we did not observe a clear dependency. We have added some more references on this and rewritten this part in order to avoid the impression that we were claiming it to have novelty. We have also removed this part from the abstract.

2) Reference 52 (now ref 70) includes data from alumina with permeation rates of 1.2x10^(-1) [at 100 nm] and 0.33x10^(-1) [at 130 nm] under pretty much the same ALD conditions and identical precursors. This manuscript shows a permeation rate of 6.9x10^(-1), that is much higher value, for a much thicker film of 200 nm! These values suggest that the permeation rates may be +/- 1 to 2 orders of magnitude, which, in turn, suggests that no significant gain, if any, is attained with the nanolaminates presented in this manuscript.

We thank the reviewer for this comment. Please note that the permeation measurements carried out by the author of the paper cited also differed from ours. In fact, in the reference 52 mentioned (now reference 70), the authors carried out a Ca test, which is based on the monitoring of corroding Ca films and carried out over a very small surface (Ca pads of 1 mm2). Please also note that in the paper mentioned, there was no PET substrate, as the ALD film was deposited on the Ca pads to encapsulate them and monitor their corrosion (these Ca pads were prepared directly on a glass substrate).

In our case, however, we carried out gas permeation testing over approximately 16 square centimeters, in order to obtain values as representative as possible. In addition, the values of the last column in Table 1 (page 10) has now been amended, as the PET contribution was taken off and the unit conversion corrected (1 Barrer = 10-10cm3(cm.sNaNHg)-1). The O2 Permeation rate of the alumina calculated for our sample was 2.3x10-1 cm3m-2day-1 ; which is still higher than the one in the reference mentioned, but to a less extent. As now indicated in the caption of Table 1, “The permeability values relate to the ceramic coatings, the PET contribution was taken off (1 Barrer = 10-10cm3(cm.sNaNHg)-1). Interfacial shear stress was calculated using the thicknesses h, without considering the thickness of the sub-surface growth.”

3) The data in Figure 8c for 2x130 nm samples seem to be in conflict with the same data included in figure 7a (i.e., saturation crack density of 50 vs. 65, respectively).

We thank the reviewer for the careful observation. This is discrepancy is due to sample to sample variation, which can be examined in detail table S2. The data in figure 8 and table 1 present the average values for all the studied samples, whereas the fragmentation curves in figure 7 are extracted from individual test pieces. Figure 7 is aimed to illustrate the measurement data used for the analysis of the mechanical parameters. The averaging to obtain the final, more accurate result is done at later stage of the data analysis. Actually due to the sample to sample variation, it is not possible to combine the crack density – strain curves. For this reason multiple test samples were measured and analyzed each individually for each material. This was now stated more clearly also in the manuscript.

4) The inclusion of standards for bulk alumina and zinc oxide in figure 1 would be helpful

We thank the reviewer for this comment. As requested, those have been added to figure 1. 

5) In figure 7a, the alumina and zinc oxide films in series can explain the results presented, when crystalline ZnO starts forming. (For example: 1/40 + 1/150 = 2/65)

We thank the review for the comment. We feel that our new SEM/TEM analysis with the subsurface growth region causing failures, explains well the differences between the fracture characteristics of alumina and ZnO.

This discussion with new data has been added to the manuscript and supplementary. 

6) Table S1 raises a few questions:

(a) the first two rows suggest .21 nm/cycle (210/1000) for zinc oxide and .20 nm/cycle (200/1000) for alumina.  The third row suggests .26 nm/cycle (250/[25+25)*10]) for the nanolaminate, which is 25-30% higher than the individual oxides; why and what does that mean for the ALD in question?

We thank the reviewer for this comment. Indeed, the growth observed is around 20-30% higher for nanolaminates. The following text and appropriate reference have been included in the manuscript to explain this phenomenon:

“The total film thickness of the compactsingle oxides was measured to around 200± 19 nm and 210± 38 nm for Al2O3 and ZnO, respectively, with relatively large point to point variations. For Al2O3 we attribute this to the sub-surface growth, see figure 2, while for ZnO the large variations could be attributed to the locally varying nanocrystalline growth. For the compactsingle Al2O3 film the growth per cycle (GPC) of 2 ± 0.2 Å at 90°C on PET is large compared to reported values of 1.3 Å to 1.4 Å for 177°C [41] and 60°C [29], respectively, on Si samples. The average ZnO the GPC of 2.1 ± 0.4 Å is only slightly larger than reported in [29] [41] but the 20% variation in our ZnO GPC is considerably larger. We attribute the larger Al2O3 GPC and the large ZnO GPC variations to the PET acting very differently than Si for ALD nucleation. We observed the total thickness of the nanolaminates to be about 20 to 30 % larger, see supplementary information S1 - S2 and table S1. This may be related to an increase in GPC for the Al2O3 layers with increasing number of ZnO layers which can be up to about 36% inferred from data by [29] at 60°C. On the other hand, the GPCs of ZnO in nanolaminates were found to reduce by 15% [29]. An effective nanolaminate GPC increase resulting from the both GPCs of Al2O3 and ZnO would thus be of about 20% which is in good agreement with the total thickness increase we observed. Similar trends for the change of GPCs of Al2O3 and ZnO at higher temperatures were also found by [41,42] and attributed (like in our case) to the higher surface area of the rough crystalline ZnO interfaces. [42]”

Ref 42: Karvonen, L., et al., Enhancement of the Third-Order Optical Nonlinearity in Zno/Al2o3 Nanolaminates Fabricated by Atomic Layer Deposition. Applied Physics Letters 2013, 103.

This comment has been added in the manuscript.

 (b) In the fourth row, the data suggests .50 nm/cycle (250/[(25+25)*10]).  Why is this almost 150% higher than the individual ALD rates?

We thank the reviewer for the careful observation. Here there seems to have been an error in the table. The correct cycle ratio was (50 + 50 )*10, and the growths per cycle for this sample are similar to the other nanolaminate samples. These were corrected in the manuscript.

(c) What is the uncertainty of all data included in Table S1?

The uncertainties for the sample thicknesses were added. The relatively large errors arise from the local thickness measurements by SEM and the variation of measurement results at different locations at the sample due to the PET substrate. Currently the nucleation of ALD films on PET seems to be not fully understood.

6) Finally, the above issues put in question the plausible mechanism presented in the later sections of the manuscript.  

With the major additions on microstructural analysis (three additional images containing TEM, and FIB cross section analysis) we strongly feel to have answered uncertainties in plausibility regarding our manuscript.

Round  2

Reviewer 1 Report

The authors have conducted further experiments and in this way gained clearer insight into microstructural factors determining the mechanical behaviour of  the nanolaminates. I recommend the article to be published as is.

Reviewer 2 Report

Publish as is. The authors have addressed all points of concern.